# Gut-Heart Axis: Microbiome Involvement in Restrictive Cardiomyopathies

**DOI:** 10.3390/biomedicines13010144

**Published:** 2025-01-09

**Authors:** Samuel Jaimez-Alvarado, Itzel Ivonn López-Tenorio, Javier Barragán-De los Santos, Dannya Coral Bello-Vega, Francisco Javier Roldán Gómez, Amedeo Amedei, Enrique Alexander Berrios-Bárcenas, María Magdalena Aguirre-García

**Affiliations:** 1Unidad de Investigación UNAM-INC, División de Investigación, Facultad de Medicina, Instituto Nacional de Cardiología Ignacio Chávez, Universidad Nacional Autónoma de México, Mexico City 14080, Mexico; samueljaal@gmail.com (S.J.-A.); itzeltenorio20@hotmail.com (I.I.L.-T.); jbarraganlos@comunidad.unam.mx (J.B.-D.l.S.); coral.vega10@gmail.com (D.C.B.-V.); 2Outpatient Care Department, Cardiomyopathy Clinic, Instituto Nacional de Cardiología Ignacio Chávez, Mexico City 14080, Mexico; roldan@cardiologia.org.mx; 3Department of Experimental and Clinical Medicine, University of Florence, 50134 Florence, Italy; amedeo.amedei@unifi.it; 4Network of Immunity in Infection, Malignancy and Autoimmunity (NIIMA), Universal Scientific Education and Research Network (USERN), 50139 Florence, Italy

**Keywords:** amyloidosis, Fabry’s disease, Wilson’s disease, cardiac systemic sclerosis, sarcoidosis, Pompe’s disease, heart failure, TMAO, prevention

## Abstract

An intriguing aspect of restrictive cardiomyopathies (RCM) is the microbiome role in the natural history of the disease. These cardiomyopathies are often difficult to diagnose and so result in significant morbidity and mortality. The human microbiome, composed of billions of microorganisms, influences various physiological and pathological processes, including cardiovascular health. Studies have shown that gut dysbiosis, an imbalance in the composition of intestinal bacteria, can contribute to systemic inflammation, a key factor in many cardiovascular conditions. An increase in gut permeability, frequently caused by dysbiosis, allows bacterial endotoxins to enter the bloodstream, activating inflammatory pathways that exacerbate cardiac dysfunction. Recent reports highlight the potential role of microbiome in amyloidogenesis, as certain bacteria produce proteins that accelerate the formation of amyloid fibrils. Concurrently, advancements in amyloidosis treatments have sparked renewed hopes, marking a promising era for managing these kinds of diseases. These findings suggest that the gut–heart axis may be a potential factor in the development and progression of cardiovascular disease like RCM, opening new paths for therapeutic intervention. The aim of this review is to provide a detailed overview of the gut–heart axis, focusing on RCM.

## 1. Introduction

### Microbial–Myocardial Axis

Cardiovascular diseases (CVD), such as coronary artery disease (CAD), stroke, heart failure (HF) and arterial hypertension, have become the leading global cause of death [1]. CVD development and prognosis is driven by well-known risk factors, including obesity, arterial hypertension, diabetes, elevated low-density lipoprotein cholesterol (LDL), smoking, and a sedentary lifestyle [2]. However, restrictive cardiomyopathies (RCM) have a genetic component and are still poorly understood. RCM are characterized by diastolic dysfunction of a non-dilated ventricle caused by abnormal tissue that replaces the normal heart muscle. The gut microbiome (GM) has been recognized as a significant modulator of systemic inflammation, metabolism, and immune responses [3] and the gut–heart axis has emerged as a key area of interest in understanding the complex relationship between GM and CVD.

A microbiome is a collection of symbiotic microorganisms. It is estimated that the human body contains between 10^13^ and 10^14^ microbial cells, with most of these microorganisms residing in the gastrointestinal tract [4]. GM performs various functions that support host health, such as fermenting indigestible dietary fibers, modulating immunity and the synthesis of vitamins, and maintaining the gut barrier [5]. Research has focused on the relationship between GM and CVD, along with its associated risk factors. Advancements in metagenomics and metabolomics have enabled the identification of specific GM metabolites that contribute to the development of various cardiovascular conditions [6], suggesting that the gut microbiome could be a risk factor for CVD.

The imbalance of GM populations (dysbiosis) alters the production of key microbial metabolites such as short-chain fatty acids (SCFAs) and trimethylamine N-oxide (TMAO) [7], which is associated with CVD risk [8,9,10]. Elevated TMAO levels have been linked to all-cause mortality in cardiac patients [8].

The GM SCFAs (produced by carbohydrate fermentation), such as acetate, propionate, and butyrate, support cellular homeostasis, acting as second messengers and providing anti-inflammatory effects [11]. This dynamic relationship highlights how gut health significantly impacts cardiovascular well-being, underscoring the crucial role of a healthy microbiome overall health (Figure 1).

Additionally, it has been demonstrated that the lipopolysaccharides (LPS) trigger signal transduction via nuclear factor kappa B (NF-κB), activating the toll-like receptor 4 (TLR4) pathway [12], leading to chronic inflammation immune responses.

CVD and gastrointestinal diseases are prevalent health issues in the general population. They are interconnected through a shared pathogenic mechanism that underlines their relationship [13]. Several gastrointestinal diseases have been linked to CVD, including rhythm disorders such as atrial fibrillation (AF), HF, and coronary atherosclerosis.

Inflammatory bowel diseases (IBD), including ulcerative colitis and Crohn’s disease, may contribute to CAD development and progression. IBD has been associated with a risk of hospitalization due to HF [14], stroke [15,16], and CAD events [15,16]. This relationship is thought to be driven by endothelial dysfunction [17].

Additionally, *Helicobacter pylori* (*H. pylori*) has been associated with CVD for its pro-atherogenic mechanisms [18], increasing LDL levels [19], disrupting glucose and lipids metabolism [20], and promoting atherosclerosis [21].

Finally, gastroesophageal reflux disease (GERD) has been related with an increased AF risk [22] and it remains an independent risk factor for developing AF [23].

The comprehension of these correlations underscores the relevance of a multidisciplinary approach to address the overlapping pathophysiology of these conditions and enhance patient outcomes. Recent evidence suggests that GM dysbiosis may be a key factor in these associations, offering new perspectives for understanding the gut–heart axis. 

## 2. Restrictive Cardiomyopathies

RCM are a heterogeneous group of cardiovascular disorders characterized by infiltrative diseases such as amyloidosis, storage disorders like hemochromatosis, and non-infiltrative diseases like systemic sclerosis and diabetic cardiomyopathy, involving the accumulation of abnormal substances within the myocardial tissue, leading to structural and functional disruption [24] (Figure 2).

In people under 30 years, RCM are primarily linked to genetic disorders causing increased fibrosis or abnormal deposition of substances like iron, proteins, or glycogen. In older adults, common causes include cardiac amyloidosis, iron overload, radiation-induced heart disease, and sarcoidosis [25]. These disorders impair the heart’s normal structure and function, which frequently results in arrhythmias like AF, thromboembolic problems, sudden cardiac death, and heart failure with preserved ejection fraction (HFpEF), which is the most prevalent manifestation [25,26].

RCM are typically diagnosed through a combination of clinical, laboratory tests, and imaging studies like echocardiography and cardiac magnetic resonance that are more valuable in identifying specific causes or narrowing down the differential diagnosis. Myocardial biopsy is a valuable but invasive diagnostic tool, typically reserved for cases with high clinical suspicion or inconclusive noninvasive test results [27].

Among RCM, cardiac amyloidosis is the most common, with a 5-year survival rate of 55% in adults, where HF accounts for 42% of fatalities [28], so timely identification is crucial for starting targeted treatments and enhancing patient prognosis. These cardiac disorders frequently provide significant insights through their clinical manifestations; in addition, diagnostic imaging and genetics studies have significantly improved the recognition and knowledge of these disorders (Table 1).

Finally, understanding the interplay between the microbiota and myocardial health could open new avenues for therapeutic strategies in these diseases.

### 2.1. Is There a Relationship Between the Gut–Liver–Heart Axis and Restrictive Cardiomyopathies?

The association between the GM and liver is a complex interplay where microbiota significantly influences liver health and disease [23]. Liver diseases such as non-alcoholic fatty liver diseases (NAFLD), non-alcoholic steatohepatitis (NASH) [29,30], autoimmune liver diseases [31], and cirrhosis [32] are strongly associated with GM dysbiosis; in addition, a significant correlation with CVD has been demonstrated. NAFLD has been linked to an increased risk of CAD, HF, stroke, and arrhythmias [33].

The GM plays an essential role in modulating bile acid (BA) metabolism and SCFA production and its dysbiosis contributes to liver inflammation and fibrosis, which may exacerbate cardiovascular dysfunction [34,35]. Understanding gut–liver–heart axis provides a framework for exploring how the GM and liver metabolism influence cardiovascular health.

Finally, cirrhotic cardiomyopathy is a well-recognized cardiac dysfunction associated with liver cirrhosis. It is characterized by impaired cardiac contractility, diastolic dysfunction, and electrophysiological abnormalities, occurring independently of preexisting cardiac diseases [36,37]. NAFLD is associated with an increased risk of cardiomyopathy, it can lead to left ventricular diastolic dysfunction and hypertrophy, contributing to congestive HF [38,39]. The GM can modulate these metabolic pathways, potentially influencing the development of cardiomyopathy through systemic inflammation and metabolic dysregulation.

The liver plays a significant role in cardiovascular health. When it comes to RCM, their relationship to liver pathology remains poorly understood and understudied. Although some evidence suggests that liver diseases may contribute to myocardial fibrosis and diastolic dysfunction, specific data linking liver diseases to the RCM development/progression are scarce. This documented data missing emphasizes the need for further research to focus the liver–heart axis in RCM. 

### 2.2. Amyloidosis

Amyloidosis is a systemic disease that covers a group of disorders characterized by the bowel movement of misfolded fibrillar proteins in the extracellular space [40,41]. Over 98% of cardiac amyloidosis are due to two main types: light chain cardiac amyloidosis (AL) and transthyretin (TTR) which at the same time is subdivided in two entities, hereditary and wild-type [42]. Hereditary-TTR amyloidosis (h-ATTR) affects younger ages, with an episode of cardiomyopathy, polyneuropathy, or mixed depending on the TTR gene mutation [41]. Meanwhile, wild-type TTR amyloidosis (wt-ATTR) most frequently affects elderly men, who begin with RCM.

Pathophysiology involves the misfolding of the tetrameric protein transthyretin, synthesized and secreted by the liver. Transthyretin is dissociated from tetramers into dimers and monomers, forming amorphous aggregates that deposit extracellularly, especially in the myocardium [43]. These deposits disrupt cardiac physiology, contributing to HFpEF, left ventricular hypertrophy, arrhythmias [44], and systemic manifestations such as carpal tunnel syndrome and lumbar canal stenosis [40,42] (Figure 3).

#### Gut Dysbiosis, Inflammation, and Protein Misfolding in Cardiac Amyloidosis

It has been reported that GM dysbiosis induces inflammation, which leads to degenerative diseases and different associated comorbidities such as amyloidosis [45]. In amyloidosis, bacterial components such as LPS and metabolites, formed by bacteria such as SCFAs, and TMAO have been associated with amyloid deposits and pro-inflammatory cytokines (e.g., interleukin-1β (IL-1β) and IL-6) [46] (Table 2). Peterle et al. conducted an experimental study observing that the serine protease protein secreted by bacteria of the phyla *Firmicutes*, specifically by the species *Bacillus subtilis*, participates in the production of amyloidogenic fragments together with TTR [46] (Table 2). Although the mechanism by which amyloidosis fibers are generated and accumulate is not entirely clear, an association between a poor prognosis of TTR amyloidosis and GM dysbiosis has been found [47]. In another study, significant differences were found in gut microbial communities between a healthy control group and AL patients that showed an abundance of *Actinobacteria* and *Verrucomicrobiota* and a decreased presence of *Bacteroidota* [48]. *Bifidobacterium*, a genus of *Actinobacteria*, has recently shown evidence of its ability to modulate host immunity by increasing immunoglobulins and inducing or reducing pro- or anti-inflammatory cytokines [48] (Table 2). Hu et al. observed in TTR amyloidosis that *Bifidobacterium* and *Eubacterium* were significantly decreased, while *Parabacteroides* was more abundant. The diminished probiotics were negatively associated with TTR cardiac amyloidosis through metabolites like GABA and taurine [49] (Table 2). Evidence proposes that GM dysbiosis may be a relevant factor in human AL progression. Additionally, in a group of mice treated with antibiotics, the authors observed that their use significantly reduces the deposits of β-amyloid protein, concluding that the changes generated in the microbiota decrease the β-amyloid protein accumulation, showing a positive change and disease improvement [25].

Notably, doxycycline, a drug used in amyloidosis treatment, modifies the gut microbiota. This alteration may influence the natural disease history, suggesting that GM could play a role in disease progression. In detail, evidence suggests that GM could trigger inflammation, protein misfolding, and tissue deposition and could have potential therapeutic targets.

### 2.3. Fabry’s Disease

Fabry disease (FD) is a rare X-linked lysosomal storage disorder caused by mutations in the GLA gene, which encodes the enzyme α-galactosidase A (α-Gal A) [63]. This enzymatic deficiency leads to the accumulation of glycosphingolipids, primarily globotriaosylceramide (Gb3), and its deacylated form, globotriaosylsphingosine (lyso-Gb3), within lysosomes across various tissues and organs [48,49,50]. This accumulation results in a multisystemic disease characterized by progressive renal failure, cardiomyopathy, and cerebrovascular complications, significantly impacting life expectancy [64,65]. Clinically, FD presents with a range of different symptoms. Cardiac involvement is common and can lead to ventricular hypertrophy, fibrosis, HF, and arrhythmias [66,67,68].

#### Metabolic Alterations Associated with Gut Dysbiosis in Fabry’s Diseases

A murine FD model demonstrated that the disease is associated with early GM compositional and functional dysbiosis. This dysbiosis is linked to alterations in fecal SCFAs levels, which may contribute to symptoms such as diarrhea and visceral hypersensitivity, as well as impaired communication along the gut–brain axis [55]. Additionally, lyso-Gb3, a metabolite accumulated in FD, has been shown to modulate the gut microbiota by increasing the biofilm-forming capacity of certain bacteria, such as *Bacteroides fragilis*, and altering the SCFAs production, especially decreasing butyrate levels [56] (Table 2). These changes in GM composition and metabolic profile could potentially exacerbate gastrointestinal symptoms in FD patients.

Furthermore, clinical observations have suggested that dietary interventions, such as a low-fermentable oligosaccharides, disaccharides, monosaccharides and polyols diet (FODMAP), may alleviate gastrointestinal symptoms in FD by reducing dysbiosis and improving gut health [69,70] (Table 2). Murine models and clinical observations underscore the significant role of GM dysbiosis in FD. This highlights the potential role of microbiota-targeted therapies in FD managing related gastrointestinal manifestations and systemic inflammation. Evidence regarding Fabry’s disease associated with this cardiac presentation is rare, probably due to a low incidence. More research is required in this regard.

### 2.4. Wilson’s Disease (Hemochromatosis)

Wilson’s disease (WD) is an autosomal recessive genetic disorder characterized by impaired copper metabolism due to mutations in the ATP7B gene, which encodes a copper-transporting ATPase [71]. This defect leads to copper accumulation primarily in the liver and subsequently in other organs, including the brain and cornea. The ATP7B protein is crucial for incorporating copper into ceruloplasmin and promoting its excretion into bile. Mutations in this protein result in defective copper excretion, leading to copper accumulation in hepatocytes, this excess generates reactive oxygen species (ROS) [71,72,73]. Cardiac involvement in WD is a recognized but less commonly discussed WD manifestation, primarily characterized by copper accumulation leading to left ventricular remodeling, hypertrophy, diastolic dysfunction, and, less frequently, systolic dysfunction [74,75].

#### Iron Metabolism and Microbial Composition in Animal Models

A study analyzed fecal iron content and microbiota in mice with mutations in the iron regulatory protein 2 (Irp2) and the hereditary hemochromatosis gene (Hfe). Irp2-/-mice showed elevated iron and other minerals in their feces compared to wild-type and Hfe-/-mice [58]. Significant differences in the bacterial populations were observed, with certain lactic acid bacteria being more abundant in specific mouse strains. Iron depletion through repeated venesection is the primary treatment for hemochromatosis, an iron-overload disorder. During treatment, increased iron absorption occurs in the gastrointestinal tract to compensate for the loss. This study shows that decreased iron availability in the colon, after venesection, leads to changes in GM composition, a phenomenon not previously studied in hemochromatosis patients [57]. These findings suggest genetic mutations in iron metabolism and changes in GM composition, which may have clinical implications for human disease progression and this myocardiopathy presentation. In addition, research in this field would bring us closer to promoting gut microbiota as a determinant of disease progression.

### 2.5. Cardiac Systemic Sclerosis

Systemic sclerosis (SS) is a complex autoimmune disorder characterized by fibrosis of the skin and internal organs, vasculopathy, and immune dysregulation [76,77]. The SS pathophysiology involves a multifaceted interplay between genetic predisposition, environmental triggers, and immunity abnormalities. This leads to endothelial cell injury, fibroblast activation, and excessive collagen deposition, resulting in tissue fibrosis and organ dysfunction [78,79].

The SS process begins with vascular injury, usually due to autoimmune attacks on endothelial cells, which leads to structural vascular changes such as capillary loss and arteriolar stenosis [80,81,82]. Myocardial fibrosis is a hallmark of cardiac involvement in cardiac systemic sclerosis (SSc), often resulting from chronic inflammation and endothelial damage. This fibrosis can lead to a range of cardiac complications including arrhythmias, HF, and conduction abnormalities [59,83,84].

#### Dysbiosis in Cardiac Systemic Sclerosis: TMAO Participation

In a study investigating the gut microbiota, adults patients completed surveys on their gastrointestinal symptoms and diet. The results showed that more severe gastrointestinal symptoms were linked to decreased microbial diversity and changes in microbial composition, with increased abundance of pathobionts like *Klebsiella* and *Enterococcus* [85] (Table 2). This points to the relevance of further research, especially through controlled trials, to better understand how diet and microbiota influence disease progression and symptoms management.

Intestinal dysbiosis is common in systemic sclerosis, as documented by [86,87], but its role in microvascular injury and fibrosis remains unclear. GM produces trimethylamine (TMA), which is converted by flavin-containing monooxygenase (FMO3) into TMAO. One study shows that TMAO reprograms skin fibroblasts, vascular endothelial cells, and adipocytic progenitor cells into myofibroblasts via the TMAO receptor protein pERK. The findings suggest that the FMO3-TMAO-pERK axis connects GM to vascular remodeling and fibrosis in SSc, offering a potential therapeutic target [60]. Other studies evidenced that the TMAO concentration is increased in SS, especially in patients with advanced organ involvement [61] (Table 2). Similarly, a recent study documented elevated TMAO circulating levels, regardless of comorbidities such as age, sex, renal function, diabetes, and CVD [62]. These findings underscore the microbiota’s influence on cardiac systemic sclerosis; given that TMAO, a metabolite produce by gut microbiota, is a potentially modifiable factor, further research is needed to explore its relationship with SS severity and progression.

### 2.6. Sarcoidosis

Sarcoidosis is an inflammatory multi-systemic disease of unknown etiology that is characterized by the formation of granulomas in different organs; pulmonary complaints are the most common [88,89,90], except for the Japanese population, in whom cardiac presentation is the most common [91]. In addition, recent studies indicate that cardiac sarcoidosis (CS) is more prevalent than previously thought [51,91].

#### Microbiota in Sarcoidosis: Immune Dysregulation and Therapeutic Prospects

The cause of sarcoidosis has not been discovered; it is believed that it involves some genetic predispositions in conjunction with environmental triggers. This rare disease can resemble or coexist with other autoimmune disorders, with different immunosuppressive treatment, supporting an autoimmune key role [92]. In addition, evidence suggests that certain microorganisms, such as *Cutibacterium acnes (C. Acnes)* and *Mycobacterium tuberculosis* may be implicated in sarcoidosis development [91,93] (Table 2), proposing microbiota as a potential therapeutic and prognosis target. Granulomatous inflammation is believed to result from a dysregulated immune response to unidentified environmental antigens in genetically predisposed individuals. CS is relatively uncommon but can present with a variety of manifestations. The most frequent symptoms include arrhythmias, such as tachycardia or heart block. Less commonly, it may lead to pericardial effusion, where fluid accumulates around the heart, or the formation of myocardial granulomas, which can result in cardiac fibrosis and impaired heart function [52,53]. These different clinical manifestations highlight the complexity involved in sarcoidosis.

The main studies associated with sarcoidosis are focused on the respiratory tract, due to its high prevalence [94]. For example, one study that researched the associations between GM and chronic respiratory diseases found three genetically predicted taxes, such as *Methanobacteria*, order *Methanobacteriales*, class *Methanobacteriaceae*, that were significantly associated with sarcoidosis [95] (Table 2). Focusing on cardiovascular systems, a study examining formalin-fixed paraffin-embedded myocardial tissues from patients with CS myocarditis or other cardiomyopathies immunohistochemistry with *C. Acnes* specific monoclonal antibody was used to identify bacterial presence. *C. acnes* was observed in 63% of cardiac sarcoidosis samples with granulomas [96] (Table 2). These findings suggest that *P. acnes* may contribute to granuloma formation in CS patients. Alternatively, a prospective, multicenter, randomized, open-label, controlled clinical trial, named JACNES in Japan, focused on evaluating the clinical response of CS patients to antibiotic therapy. This study randomly assigned patients to receive either standard corticosteroid therapy combined with antibiotics effective against *C. acnes*, or corticosteroid therapy alone [54]. The study results are currently under investigation. These findings may improve the treatment of CS patients. The GM role in sarcoidosis appears to be significant, certain microorganisms could contributegranulomatous and deposits. Dysbiosis may exacerbate disease progression by participating in cardiac and respiratory presentations. The evidence suggests that the microbiota plays a role in the pathophysiology of cardiac sarcoidosis, and its modification through the proper use of antibiotics could reduce disease progression, improving drugs that regulate the microbiota, inflammation, and immunity

### 2.7. Pompe’s Disease

Pompe disease (PD) or glycogenosis type II is a rare, chronic, and muscle-weakening, often fatal neuromuscular disease, caused by a partial or total deficiency of acid alpha-glucosidase (GAA), a key enzyme for glycogen catabolism. Glycogen, an intracellular polymer of glucose residue bonds and branching points connected by α-1,6 bonds [97]. PD has very low prevalence; there is not enough evidence related to the microbiome associated with this disease. 

#### Fecal Microbiota Transplantation in Pompe Disease: A Case Study with Gastrointestinal Infection and Dilated Cardiomyopathy

Currently, one clinical case of fecal microbiota transplantation (FMT) has been reported in an infant with PD with chronic gastrointestinal infection by *Clostridium difficile (C. difficile)* and dilated cardiomyopathy, with multiple hospitalizations and the administration of multiple doses and types of antibiotics without response to treatment, who responded favorably after FMT [98]. A 21-month-old Hispanic girl, born full-term via cesarean, was diagnosed with PD at 4 months of age. Her condition was complicated by dilated cardiomyopathy, ventilator dependence, and failure to thrive, requiring a gastrojejunostomy feeding tube. At 9 months, she developed recurrent foul-smelling diarrhea, fever, and abdominal discomfort while on antibiotics for presumed aspiration pneumonia. *C. difficile* infection was confirmed by PCR. Despite multiple treatments, she experienced recurrent *C. difficile* infection episodes with significant symptoms and hospitalizations. After 12 months of recurrent *C. difficile* infection and failed antibiotic therapies, FMT was proposed, with her mother screened as the donor. She did not experience any side effects [98]. Controlled trials found FMT to be highly effective and safe in adults, achieving up to 90% success and outperforming antibiotics [99,100]. This report highlights that FMT can be a safe and effective treatment option for the medically complex. This suggests that the microbiota may be a potential determinant for the adjuvant treatment of medical conditions in which there is little response to such treatments.

## 3. Dysbiosis: Prevention Strategies in Heart Failure

Recent evidence suggests that HF is linked to disrupted intestinal epithelial function, likely due to reduced blood flow and ischemia [101,102]. This leads to increased bowel wall thickness and gut permeability [103], resulting in the translocation of bacteria and their wall products into the circulation with inflammatory activation and modulating metabolites that can have both beneficial and harmful effects on the CVD development [103]. Harmful metabolites such as TMAO contribute to atherosclerosis, thrombosis, and are linked to cardiovascular event; on the other hand, beneficial metabolites like SCFAs can help improve blood pressure and support myocardial repair, being closely linked to dietary fiber intake.

Gut dysbiosis has been suggested as a pathogenic factor in various diseases, including HF development, supporting the “gut hypothesis” for the condition [104]. Luedde et al. observed in patients with stable HF an overgrowth of pathogenic bacteria such as *Campylobacter*, *Shigella*, *Salmonella*, *Yersinia*, and species of *Candida* [105]. In addition, there has been a reported depletion of bacteria that are known to produce SCFAs in HF patients in comparison to control subjects; in addition, HF patients were observed to have a lower intake of dietary fiber [106]. Dysbiosis is associated with HF in the reduction in bacteria that produce metabolites for cardiovascular homeostasis.

### Gut Microbiota as a Target in Heart Failure

Dietary changes, probiotics, prebiotics, and FMT [107,108] are therapeutic approaches currently used to enhance gut bacterial health; specific interventions to improve gut dysbiosis and improve cardiovascular health may still be some time away [109]. Probiotics have shown limited benefit in reducing myocardial hypertrophy in animal studies [110]; they have been linked to improved left ventricular ejection fraction [111]. The gut–heart study, currently underway, is investigating the effects of rifaximin, the probiotics yeast *Saccharomyces boulardii* (*S. boulardii*), and a no-treatment control in HF [112]. Additionally, a recent study investigated the impact of lactic-fermented bee pollen probiotics on GM [113]. The results showed an increased abundance of beneficial bacteria, *Lactobacillus* spp., and *Bifidobacterium* spp. [113], providing valuable insights into potential roles in HF management. FMT has shown success in treating gut dysbiosis but has variability in patient response [114]. Additionally, it carries a potential risk of transmitting harmful infections to the new host [115].

Indole-3-propionic acid (IPA) is a metabolite produced by GM from dietary tryptophan. IPA plays a key role in preserving mucosal homeostasis and supporting gut barrier function by binding to its receptors [116]. IPA has been reported to be reduced in patients and a mouse model [117]. IPA supplementation attenuates diastolic dysfunction, metabolic remodeling, oxidative stress, inflammation, GM dysbiosis, and gut barrier damage in a mouse HF model [117]. IPA supplementation could offer a therapeutic and prevention strategy for HF.

It has been proposed that polyphenols have beneficial effects on GM. Polyphenols and their derivatives present an opportunity to prevent and treat CVD by promoting gut eubiosis [118]. A recent study by the Optimal Nutraceutical Supplementation in Heart Failure (ONUS-HF) group confirmed the potential benefits of combining natural products, including apple-derived phlorizin, *Vitis vinifera* extracts, bergamot polyphenols, and Olea europaea derivatives, in patients at an early HF stage [119].

These findings highlight the relevance of gut health in CVD and suggest that targeting GM may become an important strategy for treating HF in the future. However, research on preventive measures and targeted interventions on GM in the context of RCM remains limited. Expanding research in this area may provide valuable insights into novel therapeutic approaches that can address unmet needs in managing complex cardiac diseases.

## 4. Drugs’ Impact on Gut Microbiota and Cardiovascular Health

GM influences drugs’ metabolism through well-established pharmacokinetic pathways, including microbial enzymes that convert drug molecules. GM interferes with pharmacokinetics and pharmacodynamics [120,121], but at the same time medication can alter GM [122,123]. Several human studies have reported an association between specific drug use and changes in microbial composition and function [122,124,125,126].

Many commonly used cardiovascular drugs showed strong interaction with GM, including, aspirin, digoxin, sodium-glucose cotransporter-2 (SGLT2), calcium channel blockers, betablockers, renin-angiotensin system inhibitors, statins, warfarin, clopidogrel, heparin, amiodarone, and antiplatelets.

Aspirin metabolism is influenced by GM [127], but its use significantly alters GM composition with *Bacteroides* and *Ruminococcaceae* [128]. GM significantly affects the digoxin bioavailability [129]. Early research showed that gastrointestinal *Eubacterium lenta* (*E. lentum*) produces an inactive metabolite of digoxin [130] and intestinal digoxin inactivation is one of the clearest associations between cardiovascular medicine and GM. Regarding the angiotensin-converting-enzyme inhibitors, a study with enalapril did not find an alteration of GM species, but reduced plasma levels of TMAO were documented [131]. These data suggest a potential enalapril role in modulating GM production of this harmful metabolite. Calcium channel blockers have demonstrated significant interactions with GM.

Amlodipine is partially metabolized by gut microbiota. In some studies, a 9% reduction in unchanged amlodipine has been observed during 24 h incubations with human faecalis, suggesting microbial involvement [132]. Beyond its pharmacological role, amlodipine exhibits antimicrobial properties, effectively inhibiting bacterial species such as *Staphylococcus aureus*, *Vibrio cholerae*, *Shigella*, and *Salmonella* [133].

Dapagliflozin is an SGLT2, used for glycemic control and blood pressure control [134] and has demonstrated its protective role in HF patients [135]. A recent animal study on HF documented that dapagliflozin treatment reduced inflammation, infarction area, and cardiac fibrosis in mice. Dapagliflozin decreased the ratio of *Firmicutes/Bacteroidetes*, which was increased in HF mice [136]. These findings suggest that dapagliflozin may modulate GM, contributing to HF treatment.

It has been reported that oral administration of vancomycin significantly impacts host GM diversity [137]. Some studies suggest that this GM population reduction in mice provided cardioprotective benefits, including smaller myocardial infarction size and lower circulating leptin levels in an ischemia/reperfusion mice model [138,139]. However, it has been reported that using an enteral-non-absorbable polymyxin B/tobramycin regimen induced decreased fecal endotoxin concentration [140]. Although antibiotics may provide some cardiovascular protection, their effects appear to be only for the treatment duration. Antibiotics often reduce the overall GM, removing harmful and beneficial bacteria.

These findings highlight the dynamic association between cardiovascular drugs, antibiotics, and GM and suggest that the modulation of the GM may become an integral part of cardiovascular treatment. 

## 5. Conclusions

Although the actual evidence reported about GM and these diseases is scarce, the evidence so far on the role of the GM in RCM have highlighted the potential impact of gut bacteria on disease progression. Understanding how dysbiosis and microbial metabolites contribute to cardiac fibrosis and HF opens up new opportunities for targeted therapies and a deeper pathophysiological knowledge of these diseases. This review approaches the emerging role of the GM in RCM pathogenesis. Amyloidosis, FD, and CS demonstrate distinct microbial imbalances: increased *Parabacteroides* and reduced *Bifidobacterium* and *Eubacterium* in amyloidosis alongside *Bacillus subtilis*, producing amyloidogenic fragments as well as altered SCFAs production linked to *Bacteroides fragilis* in FD and the participation of *C. acnes* and *Mycobacterium tuberculosis* in CS, which may contribute to granuloma formation. In conclusion, our review contributes to filling this gap by synthesizing current knowledge about RCM and GM; advancing our understanding in this area may result in the non-invasive identification of biomarkers, enabling earlier detection and GM-targeted therapies that could complement current treatments and improve their efficacy. Delving into this field represents a unique opportunity to advance personalized medicine in cardiology. 

To search for relevant information and scientific impact, a detailed search was carried out on recent reports on restrictive cardiomyopathy and its relationship with GM alterations.

We were able to corroborate that there are few clinical and scientific findings associated with intestinal dysbiosis that can indicate a relationship between the gut–heart axis; however, the obtained results suggest a close relationship between the presence of disease symptoms with GM dysbiosis and its implications in immune response.

Therefore, they lead us to discuss and investigate the field of RCM and the potential relationship it has with GM. 

There is little evidence because there appears to be a low prevalence in these associated pathologies; however, this review expands the field of research between the gut–heart axis, knowing that intestinal dysbiosis has a relevant impact, mainly due to the disruption of permeability giving way to toxins. These toxins are mobilized through the bloodstream, targeting vital organs (e.g., heart) marking an important inflammation sign and activating various mechanisms that directly impact disease prognosis. 

## 6. Perspectives

Research on the gut–heart axis offers promising opportunities to understand and treat cardiovascular diseases. Future studies should focus on identifying the fine mechanisms driving changes that influence inflammation and fibrosis in cardiovascular conditions. Metagenomics and machine learning can provide essential insights into personalized microbiota-based therapies. With precision microbiota approaches, we may uncover new therapeutic targets and preventive strategies for cardiovascular care.

## Figures and Tables

**Figure 1 biomedicines-13-00144-f001:**
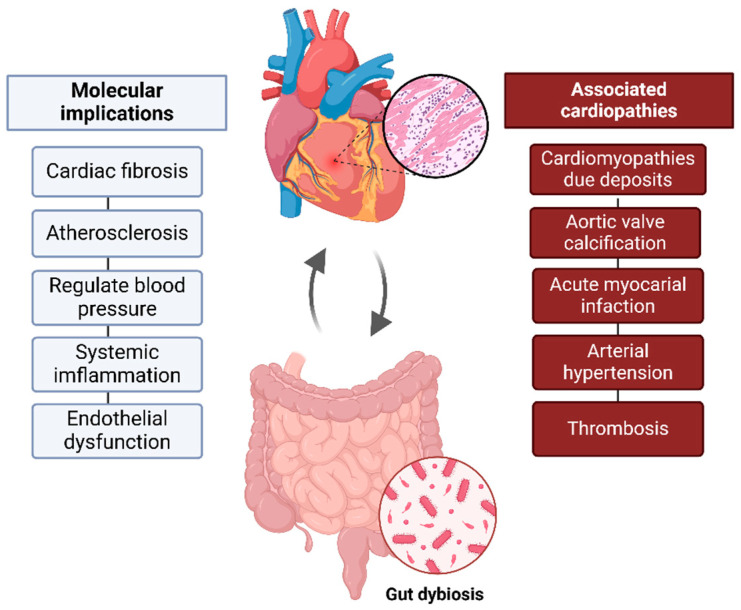
Gut–heart axis. Bidirectional connection between gut microbiota and cardiovascular health. Gut dysbiosis, can contribute to inflammation, atherosclerosis, and cardiovascular changes, influencing the development of pathological conditions such as RCM and acute myocardial infarction, among others.

**Figure 2 biomedicines-13-00144-f002:**
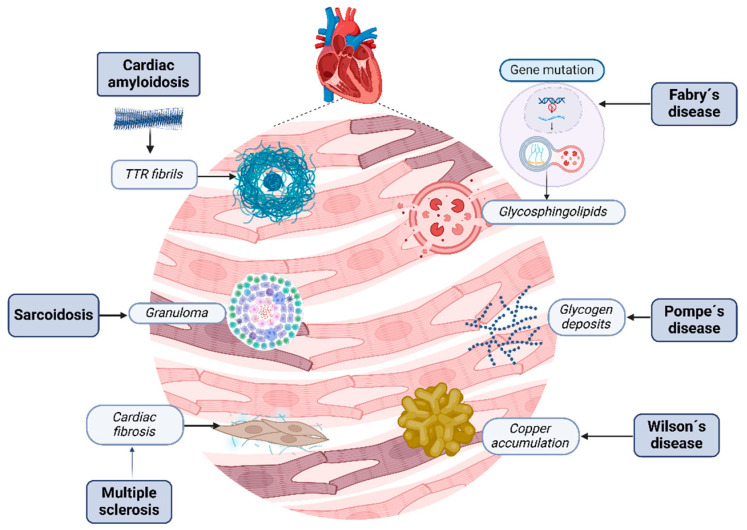
Restrictive cardiomyopathies. Several kinds of molecules and changes can contribute to heart tissue damage; each disease has its own physiopathology that induces the development of these conglomerates. Cardiac amyloidosis disrupts myocardial tissue with misfolding amyloid fibrils. Fabry’s disease is characterized by the accumulation of globotriaosylceramide due to enzymatic deficiency. Sarcoidosis is characterized by the creation of granulomas, Wilson’s disease by excessive iron in cardiomyocytes due to dysregulated iron metabolism, and, finally, Pompe’s disease by excess glycogen within lysosomes due to enzymatic deficiency. Alternatively, sclerosis is characterized by injury, fibroblast activation, and excessive collagen deposition, resulting in tissue fibrosis.

**Figure 3 biomedicines-13-00144-f003:**
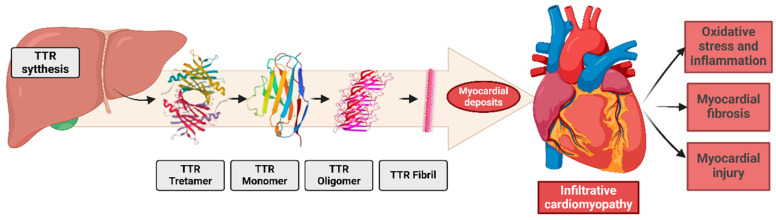
The pathogenesis of transthyretin (TTR) amyloid fibril deposition involves the misfolding and aggregation of TTR monomers into insoluble amyloid fibrils, which accumulate in the extracellular matrix. Additionally, these deposits cause molecular changes in the heart and in all cardiac systems.

**Table 1 biomedicines-13-00144-t001:** Restrictive cardiomyopathies, their etiology, infiltrative substance, principal clinical manifestations, and current treatment. Adapted and modified from [26].

Cardiomyopathies	Etiology	Infiltrative Substance	Clinical Manifestations	Treatment
Infiltrative				
Amyloidosis	Acquired/Hereditary	Amyloid	LVH, carpal tunnel syndrome	Daratumumab, tafamidis, doxycycline.
Sarcoidosis	Acquired	Granulomas	Granulomatous inflammation, arrhythmias	Corticosteroids, immunosuppressive drugs.
Storage diseases				
Fabry’s disease	Hereditary	Glycosaminoglycans	LVH, angiokeratomas	Enzyme replacement therapy.
Hemochromatosis	Hereditary	Iron	Arrhythmias, hyperpigmentation	Phlebotomy, oral chelation.
Non infiltrative				
Cardiac Systemic Sclerosis	Acquired	Collagen	Myocardial fibrosis, sclerodactyly	Corticosteroids, immunosuppressive drugs.

VA—heart valves abnormalities; LVH—left ventricular hypertrophy.

**Table 2 biomedicines-13-00144-t002:** Gut–heart axis. Findings of cardiomyopathy and its relationship with gut microbiota.

Cardiomyopathies	Model or Human Studies	Role of the Microbiota (Eubiosis/Dysbiosis)	Bacterial Association (Phyla/Species)	Reference
Amyloidosis	Experimental study -in vivo -in vitro	Dysbiosis LPS TMAO Pro-inflammatory cytokines Interleukin-1β (IL-1β) and IL-6	Phyla *Firmicutes*, species *Bacillus subtilis*	[9,46,48,49,50]
Sarcoidosis	Clinical study Experimental study -in vivo	Dysbiosis Dysregulated immune	*Methanobacteria*, order *Methanobacteriales*, class *Methanobacteriaceae* *and* bacterial presence *Cutibacteriumacnes* *Propionibacterium. acnes*	[51,52,53,54]
Fabry’s disease	Experimental study -in vivo	Dysbiosis Alterations in fecal SCFAs levels	*Bacteroides fragilis*	[55,56]
Hemochromatosis	Experimental study -in vivo	Dysbiosis Changes in GM composition	There is a lack evidence to know the bacteria involved	[57,58]
Cardiac Systemic Sclerosis	Clinical study Experimental study -in vivo	Dysbiosis decreased microbial diversity and changes in microbial composition, TMAO	Increased abundance of *Klebsiella* and *Enterococcus*	[59,60,61,62]

GM—gut microbiota; LPS—lipopolysaccharides; TMAO—trimethylamine N-oxide.

## Data Availability

Not applicable.

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
