# Peer review of "Gut-Heart Axis: Microbiome Involvement in Restrictive Cardiomyopathies"

_biomedicines, 2025, doi:10.3390/biomedicines13010144_

Round 1

Reviewer 1 Report

Comments and Suggestions for Authors

Dear Editor,

I evaluated this paper. Some points should be addressed by the authors:

- authors can consider and discuss the paper from Gesualdo M et al. J Cardiovasc Med (Hagerstown). 2016 May;17(5):330-8.

- authors should include a table gathering the main findings from the literature. This would increase the readability of the text and the comprehension of the literature background.

- the English of the paper should be revised by a native English speaker

- according to me authors should include the analysis of liver diseases for the pathogenesis of RCM in relation to microbiome. Please discuss such a point as the liver-heart axis might be involved in RCM pathogenesis and might alter the microbiome

- furthermore, the impact of medications on microbiome and the possible influence of these alterations on cardiac cells should also be discussed. This "secondary form" of cardiomyopathies might be a second focus for this interesting narrative review.

Comments on the Quality of English Language

A native English speaker should revise the paper due to several typos.

Author Response

Response to Reviewer 2

January 02, 2025

Gut-heart axis: microbiome involvement in restrictive cardiomyopathies. Manuscript ID: biomedicines-3403513

We want to thank you for the helpful comments and close attention to our manuscript provided during the review. We have carefully gone over your comments and replied to each individually.

Q1: Authors can consider and discuss the paper from Gesualdo M et al. J Cardiovasc Med (Hagerstown). 2016 May;17(5):330-8. 

A1: I would like to thank the reviewer for his point of view that is very relevant. We reviewed the manuscript as suggested.

                  CVD and gastrointestinal diseases are prevalent health issues in the general population. They are interconnected through shared pathogenic mechanism that underlines their relationship (98). Several gastrointestinal diseases have been linked to CVD, including rhythm disorders such as atrial fibrillation (AF), HF, and coronary atherosclerosis.

                  Inflammatory bowel diseases (IBD), including ulcerative colitis and Crohn´s disease, may contribute to the CAD development and progression. IBD has been associated with a risk of hospitalization due to HF (99), stroke (100,101), and CAD events (100,101). This relationship is thought to be driven by endothelial dysfunction (102).

                  Additionally, Helicobacter pylori (H. pylori) has been associated to CVD for its pro-atherogenic mechanisms (103), increasing LDL levels (104), disrupting glucose and lipids metabolism (105), and promoting atherosclerosis (106).

                  Finally, the gastroesophageal reflux disease (GERD) has been related with an increased AF risk (107) it remains an independent risk factor for developing AF (108).

The comprehension of these correlations underscores the relevance of a multidisciplinary approach to address the overlapping pathophysiology of these conditions and enhance patient outcomes. Recent evidence suggests that GM dysbiosis may be a key factor in these  associations, offering new perspectives for understanding the gut-heart axis

Please see the page 2, from lines 72 to 90.

Q2:  The authors should include a table gathering the main findings from the literature. This would increase the readability of the text and the comprehension of  the literature background.

A2: We thank the reviewer for the critical and adequate suggestion. It has been included in the end of section 2 (Restrictive cardiomyopathies), please see the page 10, line 383.

Table 2. Gut-heart axis. Findings of cardiomyopathy and its relationship with gut microbiota.

Cardiomyopathies

Model or Human studies

Role of the microbiota

(eubiosis/dysbiosis)

Bacterial association

(phyla/ species )

Reference

Amyloidosis

Experimental study

- in vivo

-in vitro

Dysbiosis

LPS

TMAO

Pro-inflammatory cytokines

Interleukin-1β (IL-1β) and IL-6

Phyla Firmicutes,

species Bacillus subtilis

Daniele Peterle et al., 2020.

Teng C., et al., 2020.

Tang WHW et al., 2019.

Yang J et al., 2022.

Hu et al., 2023.

Sarcoidosis

Clinical study

Experimental study

- in vivo

Dysbiosis

Dysregulated immune

Methanobacteria,

order Methanobacteriales, class Methanobacteriaceae

and bacterial presence

Cutibacteriumacnes

Propionibacterium. acnes

Kandolin R et al., 2016.

Ishibashi et al., 2018.

Fabry's disease

Experimental study

- in vivo

Dysbiosis

Alterations in fecal SCFAs levels

Bacteroides fragilis      

Delpetre C et al., 2023

Aguilera Corre et a., 2019

Hemochromatosis

Experimental study

- in vivo

Dysbiosis

Changes in GM composition

There is a lack evidence to know the bacteria involved

Teschke et al., 2024

Buhnik R et al., 2012.

Cardiac Systemic Sclerosis

Clinical study

Experimental study

- in vivo

Dysbiosis

decreased microbial diversity and changes in microbial composition,

TMAO

Increased abundance of Klebsiella and Enterococcus

Bonzkurt B et a., 2026.

Kim et al., 2022.

Stec et al., 2023.

Ho et al., 2024.

GM: gut microbiota; LPS: lipopolysaccharides; TMAO: trimethylamine n-oxide

Q3:  The English of the paper should be revised by a native English speaker

               A3: We thank the Reviewer for the critical observation. All the manuscript sections have been revised. 

Q4: according to me authors should include the analysis of liver diseases for the pathogenesis of RCM in relation to microbiome. Please discuss such a point as the liver-heart axis might be involved in RCM pathogenesis and might alter the microbiome

A4: We thank the reviewer for the appropriate suggestion. As suggested we added a new section:

2.1. Is there a relationship between the Gut-Liver-Heart Axis and Restrictive Cardiomyopathies?

The association between GM and liver is a complex interplay, where microbiota significantly influences liver health and disease (108). Liver diseases such as non-alcoholic fatty liver diseases (NAFLD), non-alcoholic steatohepatitis (NASH) (110,113), autoimmune liver diseases (111) and cirrhosis (112) are strongly associated with GM dysbiosis; in addition, it has been demonstrated significant correlation with CVD. NAFLD has been linked to an increased risk of CAD, HF, stroke and arrhythmias (109).

GM plays an essential role in modulating bile acid (BA) metabolism and SCFA production and its dysbiosis contributes to liver inflammation and fibrosis, which may exacerbate cardiovascular dysfunction (114,115). Understanding Gut-Liver-Heart axis provides a framework for exploring how GM and liver metabolism influence cardiovascular health.

Finally, cirrhotic cardiomyopathy is a well-recognized cardiac dysfunction associated with liver cirrhosis. It is characterized by impaired cardiac contractility, diastolic dysfunction, and electrophysiological abnormalities, occurring independently of preexisting cardiac diseases (116,117). NAFLD is associated with an increased risk of cardiomyopathy, it can lead to left ventricular diastolic dysfunction and hypertrophy, contributing to congestive HF (118,119). The GM can modulate these metabolic pathways, potentially influencing the development of cardiomyopathy through systemic inflammation and metabolic dysregulation.

Liver plays a significant role in cardiovascular health. When it comes to RCM, its relationship to liver pathology remains poorly understood and understudied. Although some evidence suggests that liver diseases may contribute to myocardial fibrosis and diastolic dysfunction, specific data linking liver diseases to the RCM development/progression are scarce. This documented data missing emphasizes the need for further research to focus the liver-heart axis in RCM.

Please see page 3 and 4, lines from 124 to 151.

Q5: furthermore, the impact of medications on microbiome and the possible influence of these alterations on cardiac cells should also be discussed. This "secondary form" of cardiomyopathies might be a second focus for this interesting narrative review.

A5: We thank the reviewer for the right suggestion, this part is therefore included in the 11 and 12 pages, lines 438 to 479.

  1. Drugs’ Impact on Gut Microbiota and Cardiovascular Health.

GM influence drugs metabolism through well-established pharmacokinetic pathways, including microbial enzymes that convert drug molecules. GM interferes with pharmacokinetics and pharmacodynamics (125,126) but at the same time medication can alter GM (120,124). Several human studies have reported association between specific drugs’ use and changes in microbial composition and function (120,121,122,123).

Many commonly used cardiovascular drugs showed strong interaction with GM, including, aspirin, digoxin, sodium-glucose cotransporter-2 (SGLT2) calcium channel blockers, betablockers, renin-angiotensin system inhibitors, statins, warfarin, clopidogrel, heparin, amiodarone, and antiplatelets.

Aspirin metabolism is influenced by GM (127) but its use significantly alters GM composition with Bacteroides and Ruminococcaceae (128). GM significantly affects the digoxin bioavailability (129). Early research showed that gastrointestinal Eubacterium lenta (E. lentum) produces an inactive metabolite of digoxin (130) and intestinal digoxin inactivation is one of the clearest associations between cardiovascular medicine and GM. Regarding the angiotensin-converting-enzyme inhibitors, a study with enalapril did not find alteration of GM species but reduced plasma levels of TMAO were documented (131). Thes data suggest the potential enalapril role in modulating GM production of this harmful metabolite. Calcium channel blockers have demonstrated significant interactions with GM.

Amlodipine is partially metabolized by gut microbiota. In some studies, have been observed a 9% reduction of unchanged amlodipine during 24-hour incubations with human faecalis, suggesting microbial involvement (132). Beyond its pharmacological role, amlodipine exhibits antimicrobial properties, effectively inhibiting bacterial species such as Staphylococcus aureus, Vibrio cholerae, Shigella, and Salmonella (133).

Dapagliflozin is a SGLT2, used for glycemic control, blood pressure control (134) and has demonstrated its protective role in HF patients (135). A recent animal study on HF documented that dapagliflozin treatment reduced inflammation, infarction area, and cardiac fibrosis in mice. Dapagliflozin decreased the ratio of Firmicutes/Bacteroidetes, which was increased in HF mice (136). These findings suggest that dapagliflozin may modulate GM, contributing to HF treatment.

It has been reported that oral administration of vancomycin significantly impacts host GM diversity (140). Some studies suggest that this GM population reduction in mice provided cardioprotective benefits, including smaller myocardial infarction size and lower circulating leptin levels in an ischemia/reperfusion mice model (138,137). However, has been reported that using enteral-non-absorbable polymyxin B/tobramycin regimen induced decreased fecal endotoxin concentration (139). Although antibiotics may provide some cardiovascular protection, their effects appear to be only for the treatment duration. Antibiotics often reduce the overall GM, removing harmful and beneficial bacteria.

These findings highlight the dynamic association between cardiovascular drugs, antibiotics and GM and suggest that modulation GM may become an integral part of cardiovascular treatment.

Finally, all suggestions are marked in the manuscript in red.

Reviewer 2 Report

Comments and Suggestions for Authors

This review article focus on Gut-heart axis: microbiome involvement in restrictive cardiomyopathies. The findings suggest that the gut-heart axis may be a potential factor in the development and progression of cardiovascular disease like RCM, opening new paths for therapeutic intervention. The aim of this review is to provide a detailed overview of the gut-heart axis, focusing on RCM. All my comments are shown in attached file and in brief herein:

 1.      Title: The title of the Ms meet the goal of the review article.

2.      Abstract:  The abstract provide a good summary of the review.

3.      Keywords: should be changed to widen the scope of the review content rather than the title.

4.      Please focus on the novelty/add value and/or importance of the subject for human health RCM prevention and strategies of treatments from GM point of view.

5.       M&M: Please indicate how literature was collected and limitation of use.

6.   The authors have to focus on the weakness and strengths in their review article.

7. Conclusion: The authors have to declare the suggested area of further research  and the beneficial and pathogenic bacteria connected with RCM.

8.  References: Plz update with 2024 reference if  any.

Comments on the Quality of English Language

The language may be improved further 

Author Response

Response to Reviewer 2

January 02, 2025

Gut-heart axis: microbiome involvement in restrictive cardiomyopathies. Manuscript ID: biomedicines-3403513

We want to thank you for the helpful comments and close attention to our manuscript provided during the review. We have carefully gone over your comments and replied to each individually.

Q1:  Title: The title of the Ms meet the goal of the review article.

A1: We thank the reviewer for the critical and appropriate suggestion. We hope to have satisfied the reviewer’s expectations.

Q2: Abstract:  The abstract provide a good summary of the review.

A2: We thank the reviewer for the critical suggestion.

Q3:  Keywords: should be changed to widen the scope of the review content rather than the title.

               A3: We thank the reviewer for the constructive suggestion. In agreement, it has already been considered. These are the chosen KeywordsAmyloidosis; Fabry´s disease; Wilson´s disease; Cardiac Systemic Sclerosis; Sarcoidosis; Pompe´s disease, Heart failure, TMAO, Prevention.Please see the page 1, lines 35 and 36. 

Q4: Please focus on the novelty/add value and/or importance of the subject for human health RCM prevention and strategies of treatments from GM point of view.

A4: We thank the reviewer for the critical and appropriate suggestion. In agreement, we have better described 3.0 section (please see pages 10-11, lines from 387 to 437).

  1. Dysbiosis: Prevention strategies in Heart Failure

Recent evidence suggests that HF is linked to disrupted intestinal epithelial function, likely due to reduced blood flow and ischemia (79,80). This leads to increased bowel wall thickness and gut permeability (81), ranging from the translocation of bacteria and their wall products into the circulation with inflammatory activation and modulating metabolites that can have both beneficial and harmful effects on the CVD development (81). Harmful metabolites such as TMAO, contribute to atherosclerosis, thrombosis and is linked to cardiovascular events, on the other hand, beneficial metabolites like SCFAs can help improve blood pressure and support myocardial repair, being closely linked to fiber dietary intake.

Gut dysbiosis has been suggested as a pathogenic factor in various diseases, including HF development and so supporting the “gut hypothesis” for the condition (82). Luedde et al. observed in patients with stable HF overgrowth of pathogenic bacteria such as Campylobacter, Shigella, Salmonella, Yersinia, and species of Candida (83). In addition, there has been reported depletion of bacteria that are known to produce SCFAs in HF patients in comparison to control subjects; in addition, HF patients were observed to have a lower intake of dietary fiber (84). Dysbiosis is associated with HF in the reduction of bacteria that produce metabolites for cardiovascular homeostasis.

3.1. Gut Microbiota as a Target in Heart Failure

Dietary changes, probiotics, prebiotics, and FMT (85,86) are therapeutic approaches currently used to enhance gut bacterial health, specific interventions to improve gut dysbiosis and improve cardiovascular health may still be some time away (87). Probiotics have shown limited benefit in reducing myocardial hypertrophy in animal studies (88), they have been linked to improved left ventricular ejection fraction (89). The Gut-Heart study, currently underway, is investigating the effects of rifaximin, the probiotics yeast Saccharomyces boulardii (S. boulardii), and no-treatment control in HF (90). Additionally, a recent study investigated the impact of lactic-fermented bee pollen probiotics on GM (97). The results showed an increased abundance of beneficial bacteria, Lactobacillus spp. and Bifidobacterium spp. (97), providing valuable insights into the potential roles in the HF management. FMT has shown success in treating gut dysbiosis but has variability in patient response (91). Additionally, it carries a potential risk of transmitting harmful infections to the new host (92).

Indole-3-propionic acid (IPA) is a metabolite produced by GM from dietary tryptophan. IPA plays a key role in preserving mucosal homeostasis and supporting gut barrier function by binding to its receptors (93). IPA has been reported to be reduced in patients and a mouse model (94). IPA supplementation attenuates diastolic dysfunction, metabolic remodeling, oxidative stress, inflammation, GM dysbiosis, and gut barrier damage in mouse HF model (94). IPA supplementation could offer a therapeutic and prevention strategy for HF.

It has been proposed that polyphenols have beneficial effects on GM. Polyphenols and their derivatives present an opportunity to prevent and treat CVD by promoting gut eubiosis (95). A recent study by the Optimal Nutraceutical Supplementation in Heart Failure (ONUS-HF) group confirmed the potential benefits of combining natural products, including apple-derived phlorizin, Vitis vinifera extracts, bergamot polyphenols, and Olea europaea derivatives, in patients at an early HF stage (96).

These findings highlight the relevance of gut health in CVD and suggest that targeting GM may become an important strategy for treating HF in the future. However, research on preventive measures and targeted interventions on GM in the context of RCM remain limited. Expanding research in this area may provide valuable insights into novel therapeutic approaches that can address unmet needs in managing complex cardiac diseases.

Q5:   M&M: Please indicate how literature was collected and limitation of use.

A5: We thank the reviewer for the appropriate suggestion

                  To search for relevant information and scientific impact, a detailed search was carried out on recent reports on restrictive cardiomyopathy and its relationship with GM alterations.                   We were able to corroborate that there are few clinical and scientific findings associated with intestinal dysbiosis that can indicate a relationship between the gut-heart axis, however, the obtained results suggest a close relationship between the presence of disease symptoms with GM dysbiosis and its implications with the immune response. Therefore, they lead us to discuss and investigate the field of restrictive cardiomyopathy and the potential relationship it has with the gut microbiota

It was included in conclusion section, page 13, lines 499 to 507.

Q6: The authors have to focus on the weakness and strengths in their review article.

A6: We thank the reviewer for the right suggestion

                  There is little evidence because the pathology can be indicated with a low prevalence in these associated pathologies; however, this review expands the field of research between the gut- heart axis, knowing that intestinal dysbiosis has a relevant impact mainly due to the disruption of permeability giving way to toxins. These last are mobilized through the bloodstream, targeting vital organs (e.g. heart) marking as an important inflammation sign and activating various mechanisms that directly impact on the disease prognosis.

Please see page 13, lines from 508 to 514.

Q7: Conclusion: The authors have to declare the suggested area of further research and the beneficial and pathogenic bacteria connected with RCM.

A7:  We thank the reviewer for the critical and appropriate suggestion. In agreement, we have better described conclusion section (please see the lines from page 12-13, lines 486 to 499).

Understanding how dysbiosis and microbial metabolites contribute to cardiac fibrosis and HF opens new opportunities for targeted therapies and a deeper pathophysiology knowledge of these diseases. This review approaches the emerging GM role in the RCM pathogenesis. Amyloidosis, FD and CS demonstrate distinct microbial imbalances, increased Parabacteroides, and reduced Bifidobacterium and Eubacterium in amyloidosis alongside Bacillus subtilis producing amyloidogenic fragments; altered SCFAs production linked to Bacteroides fragilis in FD, and the participation of C. acnes and Mycobacterium tuberculosis in CS which may contribute to granuloma formation. In conclusion, our review contributes to filling this gap by synthesizing current knowledge about RCM and GM, advancing us understanding in this area may serve as non-invasive biomarkers, enabling earlier detection and GM-targeted therapies could complement current treatments improving their efficacy. Delving into this field represents a unique opportunity to advance personalized medicine in cardiology.

Q8: References: Plz update with 2024 reference if any.

A8: In agreement with the right suggestion.  We have reviewed the references, updated them and added some more. Therefore, it has been added to the references section.

Finally, all suggestions are marked in the manuscript in red.

Round 2

Reviewer 1 Report

Comments and Suggestions for Authors

authors well addressed previous comments. The paper improved very much